# Sex Differences in Liver, Adipose Tissue, and Muscle Transcriptional Response to Fasting and Refeeding in Mice

**DOI:** 10.3390/cells8121529

**Published:** 2019-11-27

**Authors:** Nadezhda Bazhan, Tatiana Jakovleva, Natalia Feofanova, Elena Denisova, Anastasia Dubinina, Natalia Sitnikova, Elena Makarova

**Affiliations:** 1The Laboratory of Physiological Genetics, The Institute of Cytology and Genetics, 630090 Novosibirsk, Russia; tatyanajakovleva@yandex.ru (T.J.); nataly.feofanova@gmail.com (N.F.); elena_nsib@list.ru (E.D.); dubinina_anastas@mail.ru (A.D.); enmakarova@gmail.com (E.M.); 2Department of Physiology, Novosibirsk State University, 630090 Novosibirsk, Russia; kiara112@mail.ru

**Keywords:** sex-specific *Fgf21* gene expression, lipid glucose oxidation, fasting refeeding, mice, liver, adipose tissues, muscle

## Abstract

Fasting is often used for obesity correction but the “refeeding syndrome” limits its efficiency, and molecular mechanisms underlying metabolic response to different food availability are under investigation. Sex was shown to affect hormonal and metabolic reactions to fasting/refeeding. The aim of this study was to evaluate hormonal and transcriptional responses to fasting and refeeding in male and female C57Bl/6J mice. Sex asymmetry was observed both at the hormonal and transcriptional levels. Fasting (24 h) induced increase in hepatic *Fgf21* gene expression, which was associated with elevation of plasma FGF21 and adiponectin levels, and the upregulation of expression of hepatic (*Pparα*, *Cpt1α*) and muscle (*Cpt1β, Ucp3*) genes involved in fatty acid oxidation. These changes were more pronounced in females. Refeeding (6 h) evoked hyperinsulinemia and increased hepatic expression of gene related to lipogenesis (*Fasn*) only in males and hyperleptinemia and increase in *Fgf21* gene expression in muscles and adipose tissues only in females. The results suggest that in mice, one of the molecular mechanisms underlying sex asymmetry in hepatic *Pparα*, *Cpt1α*, muscle *Cpt1β*, and *Ucp3* expression during fasting is hepatic *Fgf21* expression, and the reason for sex asymmetry in hepatic *Fasn* expression during refeeding is male-specific hyperinsulinemia.

## 1. Introduction 

The prevalence of obesity is increasing worldwide. Food restriction is known to be one of the main approaches to correct obesity. Effective use of fasting is limited by the “refeeding syndrome”, which is expressed as increased food intake after hunger and rapid weight gain due to the increase of white adipose tissue (WAT) mass. Fundamental aspects of metabolic homeostasis are known to be regulated differently in males and females [1,2,3,4]. There is increasing evidence that sex hormones regulate the expression of genes and proteins involved in lipid and glucose turnover [5]. Moreover, thousands of genes show sexual dimorphism in the liver [6], adipose tissues [5,7], and muscle [5]. 

A number of hormones and metabolic molecules are known to regulate adaptation to fasting. Fibroblast growth factor 21 (FGF21), discovered nearly two decades ago, has been added to the list of factors that regulate organism response to food deprivation [8]. FGF21, a hormone secreted by the liver, was first discovered as a metabolic regulator that has beneficial metabolic effects on insulin resistance and diabetes [8]. A growing body of research suggests that FGF21 also plays an important role in the maintenance of energy homeostasis in several stressful conditions, including nutrient starvation [9,10,11,12]. 

In starvation, the plasma FGF21 level increases due to hepatic FGF21 expression, which is induced via the peroxisome proliferator-activated receptor α (PPARα) [10]. FGF21 stimulates lipolysis in white adipose tissue and ketogenesis in the liver. Hepatic FGF21 induction contributes to the alleviation of fasting-induced hepatosteatosis by enhancing the expression of the genes involved in fatty acid oxidation [10,12]. The effects of FGF21 are partially realized through the regulation of the expression of genes involved in carbohydrate-lipid metabolism in the liver, white visceral adipose tissue (VAT), brown adipose tissue (BAT), and muscle [11,13,14,15]. 

Previously it was demonstrated that the fasting-induced increase in hepatic *Fgf21* gene expression and circulating FGF21 levels [16,17], as well as refeeding-induced increase in VAT and BAT *Fgf21* gene expression, were biased toward females [16]. Data on the effect of FGF21 on its target gene expressions were obtained in experiments with the transgenic activation/suppression of *Fgf21* gene expression in the liver and adipose tissues [9,12] or in pharmacological experiments with recombinant FGF21 administration [8,13]. It is unknown whether sex asymmetry in FGF21 response to fasting/refeeding is associated with the asymmetry of its target gene expressions in the liver, adipose tissues, and muscle. 

Many studies performed on humans and rodents have demonstrated the impacts of sex on hormonal-metabolic reaction to fasting/refeeding. Benz et al. [7] found that caloric restriction resulted in a greater relative reduction of total and gonadal fat mass, increased lipolytic activity, augmented lipid-oxidation, and increased expressions of enzymes involved in lipolysis (adipose triglyceride lipase, ATGL and hormone sensitive lipase, HSL) in female compared to male mice. Sex differences were also described in response to refeeding and feeding. Refeeding-induced increases in circulating leptin concentrations were higher in female than in male mice [18], and postprandial increases in plasma triglyceride-rich lipoprotein, insulin, and free fatty acid (FFA) levels were found to be less pronounced in women compared with men [19]. The transcriptional mechanisms involved in this sex-dependent adaptation of energy balance remain unclear. The aim of this study was to evaluate sex-dependent hormonal and transcriptional mechanisms underlying adaptation to fasting and refeeding in mice. 

Fatty acid oxidation, ketogenesis, gluconeogenesis in the liver, lipolysis of triglycerides in white adipose tissue, thermogenesis in brown adipose tissue, fatty acid, and glucose oxidation in extrahepatic metabolic organs (WAT, BAT, muscles) are the main processes involved in adaptation to the fasting–refeeding cycle. We have focused on the mRNA levels of *Fgf21* and ligand-activated transcription factors (PPARalpha and PPARgamma) involved in regulation of FGF21 expression [10,20]. Among the genes related to lipid metabolism, we measured the expression of genes involved in the fatty acid oxidation (peroxisome proliferator activated receptor γ coactivator protein-1α, *Ppargc1a*, carnitine palmitoyltransferase 1a *Cpt1α*, and uncoupling protein 3, *Ucp3*) and also genes controlling lipolysis (hormone-sensitive lipase, *Lipe*) and lipogenesis (fatty acid synthase, *Fas,* and lipoprotein lipase, *Lpl,* the enzyme that hydrolyzes blood triglycerides and promotes the cellular uptake of free fatty acids [21]). In WAT, PPARγ, and glucose transporter 4 (*Slc2a4*), genes are also considered as lipogenic genes [22]. Among the genes related to glucose turnover, we measured hepatic expression of gluconeogenic genes (glucose-6-phosphatase, *G6p*, and phosphoenolpyruvate carboxykinase 1, *Pck1*), genes involved in glucose oxidation (glucokinase, *Gck,* and pyruvate kinase, *Pklr*), and glucose transporter 2, *Slc2a2*. In adipose tissues and muscle, we measured mRNA level of *Slc2a1*, *Slc2a4*, and insulin receptor gene, *Insr*, the latter only in the muscle. Among thermogenic genes, we measured expression of uncoupling protein 1, *Ucp1*, and deiodinase iodothyronine 2, *Dio2*, the latter only in the brown adipose tissue.

Our study demonstrated sex asymmetry both in the hormonal and transcriptional response to nutritional contrasting conditions. Sex differences were observed in elevation of plasma FGF21 and adiponectin levels during fasting and leptin and insulin levels during refeeding. Fasting-induced female-biased increase in *Fgf21* gene expression in the liver and the circulating FGF21 level were associated with the upregulation of mRNA level of genes involved in lipid oxidation in the liver (*Cpt1a*) and muscle (*Cpt1b*, *Ucp3*). Refeeding-induced male-specific hyperinsulinemia was accompanied by elevation of liver mRNA level of *Fasn* that regulates the rate of lipogenesis. These results suggest that insulin, leptin, and FGF21 are involved in the formation of sex differences in the transcriptional response to fasting and refeeding in mice. 

## 2. Materials and Methods

### 2.1. Animals

All experiments were performed according to the ‘European Convention for the Protection of Vertebrate Animals used for Experimental and other Scientific Purposes’ (Council of Europe No 123, Strasbourg 1985) and Russian national instructions for the care and use of laboratory animals. The protocols were approved by the Independent Ethics Committee of the Institute of Cytology and Genetics (Siberian Division, Russian Academy of Sciences, protocol No 35 of 26 October 2016).

C57BL/6J mice were bred in the vivarium of the Institute of Cytology and Genetics. The mice were housed individually for 3 weeks before the start of the experiment under a 12 h:12 h light:dark regime at an ambient temperature of 22 °C. They were provided ad libitum access to commercial mouse chow (Assortiment Agro, Turakovo Village, Moscow oblast, Russia) and water. Fifteen-week-old female and male mice weighing 26.8 ± 0.3 g (males) and 22.5 ± 0.5 g (females) were used.

### 2.2. Study Design

Both male and female mice were divided into three experimental groups. The first group was composed of control mice that consumed standard laboratory chow; the second group was composed of fasted mice that were deprived off food for 24 h; the third group comprised mice that consumed food for 6 h after 24 h of fasting (refed mice). Animals were deprived off food at 9:00 am. The mice from the second group were decapitated after 24 h of food deprivation at the same time as the control mice. The mice from the third group were killed by decapitation after 6 h of refeeding. To measure food intake, preweighed chow was placed at 9:00 am to mice from control and refeeding groups and weighed again after 24 h in control mice and after 6 h in refed mice. Food intake was calculated by subtracting the remaining chow and food spillages from its initial weight. The SAT, VAT, BAT, and liver mass indexes were calculated as the ratio of tissue mass to body weight expressed as a percentage. There were 8–10 mice in each experimental group. 

Trunk blood was collected after decapitation to measure hormone and metabolite concentrations. Liver, total visceral adipose tissue (VAT), total subcutaneous adipose tissue (SAT), and interscapular brown adipose tissue (BAT) were weighed. Gene expression was measured in the samples of the liver, VAT (paragonadal location), BAT, SAT (inguinal location), and skeletal muscle *Musculus quadriceps femoris*. In mice, musculus quadriceps femoris is characterized by combination of two metabolic pathways: anaerobic glycolysis and aerobic beta-oxidation, which closely interact with each other [23]. Therefore, the transcription of genes involved both in glycolysis and in the fatty acid oxidation was measured. Gene expression was measured in 6–7 mice from each experimental group.

### 2.3. Plasma Assays

Concentrations of FGF21, insulin, adiponectin, and leptin were measured using the following ELISA Kits: Rat/Mouse Fibroblast Growth Factor-21 ELISA Kit (cat. # EZRMFGF21-26K; EMD Millipore St. Louis, MI, USA; intra-assay: <10%, inter-assay: <8%); Rat/Mouse Insulin ELISA Kit (cat. # EZRMI-13K; EMD Millipore, St. Louis, MI, USA; inter-assay: 6.0% to 17.9%, intra-assay: 0.9% to 8.4%); Mouse Adiponectin ELISA Kit (cat. # EZMADP-60K; EMD Millipore, St. Louis, MA, USA; inter-assay: 1.4% to 10.8%, intra-assay: 3.8% to 8.2%); and Mouse Leptin ELISA Kit (cat. # EZML-82K; EMD Millipore, St. Louis, MI, USA; inter-assay: 3.0% to 4.6%, intra-assay: 1.1% to 1.8%). Concentrations of glucose, FFA, triglycerides, and cholesterol were measured colorimetrically using Fluitest GLU (cat. # 4342; Analyticon Biotechnologies AG, Lichtenfels, Germany; inter-assay: 1.46% to 1.88%, intra-assay: 0.45% to 0.67%); NEFA FS (non-esterified fatty acids) (cat. # 1 5781 99 10 935; DiaSys Diagnostic Systems GmbH, Holzheim, Germany; inter-assay: 1.07% to 1.15%, intra-assay: 0.98% to 1.07%); Fluitest TG (cat. # 5741; Analyticon Biotechnologies AG, Lichtenfels, Germany; inter-assay: 1.14% to 1.62%, intra-assay: 0.70% to 1.19%), and Fluitest CHOL (cat. #4241; Analyticon Biotechnologies AG, Lichtenfels, Germany; inter-assay: 1.06% to 1.93%, intra-assay: 0.55% to 1.32%). 

### 2.4. Relative Quantitation Real-Time PCR

Total RNA was isolated from tissue samples with ExtractRNA (Evrogen, Moscow, Russia) according to the manufacturer’s instructions. First-strand cDNA was synthesized with Moloney murine leukemia virus (MMLV) reverse transcriptase (Evrogen, Moscow, Russia) and oligo(dT) as a primer. 

Applied Biosystems TaqMan Gene Expression Assays, listed in Table 1, and qPCRmix-HS LowROX Master Mix (Evrogen, Moscow, Russia) were used for relative quantitation real-time PCR with β-actin as an endogenous control. Sequence amplification and fluorescence detection were performed with the Applied Biosystems ViiA™ 7 Real-Time PCR System (Life Technologies, 5791 Van Allen Way, Carlsbad, CA, USA). Relative quantitation was performed by the comparative CT method, where CT is the cycle threshold.

### 2.5. Statistical Analysis

The results are presented as means ± SE from the indicated number of mice. Two-way ANOVA was used with sex and the experimental group (control, fasting, refeeding) as factors. Differences between means were determined by post hoc Fisher’s Least Significant Difference (LSD) test. Where indicated, groups were also compared using Student’s *t*-test. Significance was determined as *p* < 0.05. The STATISTICA 6 software package (StatSoft, TIBCO Software Inc., 3307 Hillview Avenue, Palo Alto, CA 94304, USA) was used for analysis.

## 3. Results

### 3.1. Food Intake, Body Weight, Tissue Weights, and Blood Biochemistry 

Sex had no effect on food intake in either control or refed mice. Control males that consumed food for over 24 h had an average intake of 3.5 ± 0.2 g, (*n* = 9), and females had an average intake of 3.4 ± 0.1 g (*n* = 9). Refed males that consumed food over 6 h after fasting had an average intake of 1.9 ± 0.1 g (*n* = 9) and refed females had an average intake of 1.7 ± 0.2 g (*n* = 9). Sex affected the weight of body (*p* < 0.001) and the absolute weights of liver (*p* < 0.001) and SAT (*p* < 0.001) (Figure 1) but did not affect the absolute weights of VAT or BAT. In all experimental groups, the female body weights were lower, and SAT weights were approximately two times higher than those of males (Figure 1). Fasting significantly decreased and refeeding reversed the weights of body (*p* < 0.01), liver (*p* < 0.0001), and BAT (*p* < 0.01) in both males and females. There were no sex differences in the weights of VAT and BAT in control groups, and fasting significantly decreased BAT weight in both males and females, VAT weight only in males, and SAT weight only in females (Figure 1). There were no differences in organ and body weights in refed and control animals, except for VAT weights in males which did not increase during refeeding and remained lower than in control males (Figure 1). Thus, changes induced by fasting/refeeding were more pronounced in SAT of female mice and in the VAT of male mice. Changes in the relative weights of organs reflected the pattern of changes in their absolute weights (Table 2): sex affected the relative weight of SAT (it was higher in females), and fasting/refeeding affected the relative weight of the liver and SAT. BAT relative weight was higher in females. 

Neither sex nor fasting/refeeding affected free fatty acid (FFA) or triglyceride (TG) plasma concentrations (Figure 2). Fasting decreased and refeeding normalized plasma glucose levels similarly in males and females (*p* < 0.01 for the “experimental group” factor). Both sex (*p* < 0.001) and fasting/refeeding (*p* < 0.05) affected plasma concentrations of adiponectin: these concentrations increased in fasted and decreased in refed mice and were higher in females than in males in all experimental groups. Fasting also sharply increased and refeeding decreased circulating FGF21 levels (*p* < 0.05). Fasting-induced changes in FGF21, and adiponectin concentrations were more pronounced in females. Two-way ANOVA revealed effects of the factors “sex” and “experimental group” and factor “interaction” on the levels of insulin and leptin (*p* < 0.05 for “sex”, *p* < 0.001 for “experimental group”, and *p* < 0.05 for factor interaction in both cases). Fasting decreased and refeeding increased the levels of insulin and leptin in male and female mice. However, there were no sex differences in plasma insulin or leptin concentrations in control and fasted mice. Sex differences were observed only in refed mice: plasma leptin concentrations in refed females were 1.4 times higher and insulin concentrations 3.3 times lower than in refed males.

### 3.2. Liver Gene Expression

ANOVA revealed a statistically significant effect of the experimental group for hepatic genes involved in fatty acid oxidation (*p* < 0.01 for *Ppara* and *p* < 0.001 for *Ppargc1a*, *Cpt1a*, and *Fgf21*), gluconeogenesis (*p* < 0.01 for *G6pc and Pck1*), and glucose oxidation (*Gck, p* < 0.01). Fasting upregulated and refeeding downregulated the expression of the genes related to fatty acid oxidation and gluconeogenesis both in males and females. *Gck* gene expression was downregulated by fasting and upregulated by refeeding also regardless of mouse sex (Figure 3). 

Sex affected the transcriptional response to fasting/refeeding of *Ppara* (*p* < 0.05), *Cpt1a* (at the level of tendency *p* < 0.09), and *Slc2a2* (*p* < 0.05): The mRNA levels of these genes were higher in females than in males in all experimental groups. ANOVA also revealed factor interaction for transcriptional responses of *Fgf21* and *Fasn* (*p* < 0.05 in both cases). *Fgf21* expression was equal in control and refed mice regardless of sex, and fasting caused an eightfold increase in *Fgf21* mRNA levels in males and eighteen-fold increase in females. As a result, sex-differences in *Fgf21* expression were observed only in fasted mice. Fasting decreased *Fasn* expression both in males and females (*p* < 0.05, two-way ANOVA with “control” and “fasting” levels for the factor “experimental group”), and there were no sex differences in the *Fasn* mRNA levels in control and fasted mice. Refeeding caused a statistically significant fourteen-fold increase in *Fasn* mRNA levels only in males. In refed females, the *Fasn* mRNA levels were about ten times lower than in refed males.

### 3.3. Adipose Tissue Gene Expression

In SAT, *Fgf21* expression was undetectable. Food availability did not affect the expression of most SAT genes with the exception of the *Pparg* gene. Fasting downregulated and refeeding restored *Pparg* mRNA levels (*p* < 0.05) (Figure 4). The response was less pronounced in females than in males. The expression of four genes measured in SAT demonstrated sex polymorphism: mRNA levels of genes related to adipogenesis (*Pparg*), lipolysis *(Lipe*), and lipogenesis (*Lpl,*) were lower (*p* < 0.05, *p* < 0.01, and *p* < 0.05, respectively), and mRNA level of the *Cpt1a* gene, involved in β-oxidation, tended to be higher (*p* < 0.06) in females than in males (Figure 4).

In VAT, the transcription profiles of *Pparg, Ppargc1a,* and *Slc2a4* genes resembled those in SAT (Figure 5). Fasting/refeeding affected the expression of four genes: *Pparg* (*p* < 0.05), *Cpt1a* (*p* < 0.05), *Fgf21* (*p* < 0.01), and *Slc2a4* (*p* < 0.01). Fasting downregulated and refeeding normalized the expression of the *Pparg* and *Slc2a4* genes in both males and females. As in SAT, sex affected *Pparg* expression (*p* < 0.05): The mRNA levels of this gene were lower in females than in males. ANOVA revealed factor interaction for the *Fgf21* (*p* < 0.05) and *Cpt1a* (at the level of tendency *p* < 0.07) genes. In contrast to that in the liver, the VAT *Fgf21* mRNA level did not increase after fasting but did after refeeding. The refeeding-induced *Fgf21* gene upregulation was biased toward females. Only in males, fasting affected the mRNA level of the *Cpt1a* gene.

In BAT, sex had no effect on the mRNA levels of the measured genes (Figure 6). Fasting/refeeding affected the expressions of three genes involved in lipid oxidation: *Ppara* (*p* < 0.001), *Ppargc1a* (*p* < 0.01), and *Fgf21* (*p* < 0.01). mRNA levels of *Ppara* and *Ppargc1a* were sharply downregulated by refeeding regardless of sex. Refeeding-induced *Fgf21* gene upregulation was biased toward females (Figure 6).

### 3.4. Muscle Gene Expression

In muscle tissue, experimental conditions affected the expression of three genes: *Ucp3, Insr, and Fgf21* (*p* < 0.05 in all cases) (Figure 7). Fasting equally enhanced muscle *Insr* expression in males and females. Fasting upregulated and refeeding downregulated the expression of *Ucp3*, an indicator of fatty acid oxidation. These effects were more pronounced in females. Although ANOVA did not reveal influence of experimental conditions on the expressions of *Cpt1b*, marker of beta-oxidation, and *Slc2a4,* marker of insulin sensitivity, the use of the Student’s *t* test showed that fasting increased expressions of these genes (*p* < 0.05, Student’s *t* test for both genes) only in females (*Cpt1b*: 0.33 ± 0.12 (*n* = 7) vs. 1.43 ± 0.43 (*n* = 7); *Slc2a4*: 0.47 ± 0.14 (*n* = 7) vs. 1.45 ± 0.39 (*n* = 7)). *Fgf21* mRNA levels were undetectable in fasted mice, so we compared *Fgf21* expression in the refed and control groups. The transcriptional responses of the *Fgf21* gene were sex-dependent (interaction of sex x experimental group, *p* < 0.05): Refeeding-induced increase in *Fgf21* mRNA levels was found only in females. In refed females, *Fgf21* mRNA levels were 38 times higher than in control females and 6.3 times higher than in refed males.

## 4. Discussion

The aim of this study was to evaluate sex-dependent hormonal and transcriptional mechanisms underlying adaptation to fasting and refeeding in mice. We evaluated metabolic response to fasting and subsequent refeeding in male and female mice using morphometric and hormonal parameters and expression levels of some genes involved in the regulation of glucose and lipid metabolism in liver, muscles, BAT, and WAT of different localizations.

In general, males and females equally responded to the corresponding metabolic situation: fasting caused decrease of body and liver weights, loss of fat stores, hypoglycemia, decrease of leptin and increase of FGF21 and adiponectin blood levels. Refeeding caused opposite changes: Increase of body and liver weights, normalization of blood glucose levels, increase of insulin and leptin levels, decrease of FGF21 and adiponectin levels. 

Liver expression of genes involved in glucose and lipid metabolism also changed in the same way in males and females, demonstrating activation of gluconeogenesis (increased expression of the genes for glucose-6-phosphatase (*G6pc*) and phosphoenolpyruvate carboxykinase (*Pck1*)), activation of fatty acid oxidation (increased expression of the gene for carnitine palmitoyltransferase I (*Cpt1*)), inhibition of glycolysis (decreased gene expression of glucokinase (*Gck*)) and lipogenesis (decreased gene expression of fatty acid synthase (*Fasn*)) during fasting, and the opposite changes during refeeding: inhibition of gluconeogenesis and beta oxidation (lowering of *G6pc*, *Pck1*, and *Cpt1* expression) and activation of lipogenesis and glycolysis (elevation of *Fasn* and *Gck* expression). The expression of genes encoding regulatory proteins also changed in the same way in males and females: the expression of *Ppara*, *Fgf21*, and *Ppargc1a*, which are assumed to be activators of gluconeogenesis and fatty acid oxidation [11,24], increased during fasting and decreased after refeeding. 

Sex differences were observed in the severity of the hormonal and transcriptional responses to fasting and refeeding. In response to fasting, FGF21 and adiponectin blood levels, liver *Fgf21* and *Cpt1a* expression, as well as *Cpt1b* and *Ucp3* muscle expression, were elevated to a greater extent in females than in males. As liver is the main source of FGF21 in blood, the FGF21 blood level directly depends on *Fgf21* liver expression [9,11], and changes in blood FGF21 concentrations reflect the rate of liver *Fgf21* transcription. The more pronounced enhancement of adiponectin blood levels in females during fasting is probably due to the FGF21-dependent activation of adiponectin synthesis and secretion by adipose tissue [25]. In turn, adiponectin increases fatty acid oxidation in muscles, both adiponectin and FGF21 activate mitochondrial biogenesis and fatty acid oxidation in liver [25], and their combined effect on these processes may be one of the causes of more intense expression of *Cpt1a* in liver and of *Cpt1b* and *Ucp3* in muscles of females during fasting. The identified sex differences in the fasting-induced expression of genes encoding enzymes that determine beta-oxidation rate in mitochondria suggest that liver and muscle fatty acid oxidation during fasting is more intense in females; however, this assumption should be verified by biochemical methods. 

Thus, fasting-induced liver *Fgf21* expression can trigger sex differences in response to fasting via mechanisms that regulate expression of genes related to fatty acid oxidation in the liver and muscles. The increased liver *Fgf21* expression in fasting females compared to males was detected previously [16,17]. It is probably caused by sex steroids. Recently, it was shown that in C57Bl mice 40-h fasting caused more pronounced increase in FGF21 blood level in females than in males; ovariectomy eliminated this difference, and administration of estradiol at physiological doses restored it [17]. In females, the increased liver expression of *Ppara* during fasting can also contribute to sex differences in *Fgf21* expression, since PPARa is a key regulator of *Fgf21* expression in liver [9,10,26], and in females, *Ppara* expression was higher than in males. The causes of sex differences in *Ppara* expression remain unclear: Although ovariectomy reduced liver expression of *Ppara* in rats and estradiol administration restored it [27], in mice, ovariectomy followed by restoration of estradiol levels did not affect the *Ppara* mRNA level in liver [28].

We also found sex differences in the mass of subcutaneous fat and in the fasting-induced loss of fat in VAT and SAT. These results are in line with the data obtained in mice by other authors [29,30]. Sex differences in SAT weight arose regardless of feeding regime. In humans, preferential subcutaneous fat deposition was also described in women [see for rev 2]. The predominant SAT accumulation in females is obviously associated with estradiol, which increases antilipolytic α2-adrenergic receptors only in SAT but not in abdominal WAT [31]. In addition, *Lipe* mRNA levels in subcutaneous adipose tissue in females were lower than in males regardless of their nutritional status. The decreased *Lipe* expression in females compared to males may indicate lower lipolysis rate in subcutaneous fat in females and may be one of the causes of higher weight of subcutaneous fat in females than in males.

The sex-specific response to fasting from adipose tissue of different localizations was accompanied by sex-dependent differences in the expression of the *Cpt1a* gene in the visceral and subcutaneous fat. In males, unlike females, fasting activated *Cpt1a* expression in VAT, and the *Cpt1a* mRNA level during fasting became higher than in females. In subcutaneous fat, in contrast, the *Cpt1a* mRNA level was higher in females than in males. It is possible that during fasting, the fat oxidation rate in subcutaneous fat is higher in females, whereas in visceral fat, it is higher in males. These differences can contribute to sex differences in the loss of adipose tissue of different localizations during fasting. 

Sex differences common for subcutaneous and visceral fat were observed for *Pparg* expression: *Pparg* mRNA level was higher in males, and its decrease in response to fasting in males was more pronounced than in females. Other authors [32] also noted a decrease in *Pparg* expression in adipocytes in response to fasting. Sex differences in *Pparg* mRNA were probably due to sex steroid actions. Estradiol administration to ovariectomized mice was shown to decrease the level of *Pparg* mRNA in adipose tissue [33].

In response to refeeding, the blood levels of hormones changed unidirectionally in males and females, but the severity of the reaction depended on sex. After 6 h of refeeding, FGF21 and adiponectin levels decreased to normal values in mice of both sexes. Since both hormones increased to a greater extent during fasting in females, the decrease in their concentrations during refeeding in females was more pronounced. The responses from insulin and leptin differed significantly in males and females: In males, refeeding was accompanied by hyperinsulinemia, while leptin increased to normal values; in females, on the contrary, insulin increased to normal values, and an increase in leptin levels led to hyperleptinemia. As a result, in refed mice, insulin levels were higher, and leptin levels were lower in males compared to females. A similar sex-specific response of insulin and leptin to refeeding in mice has been reported by other authors [18,34]. It is possible that increased levels of leptin in the blood of females were associated with the action of sex hormones. It has been demonstrated that testosterone reduces the concentration of leptin [35], and estrogen increases leptin production in vivo [36] and in vitro [37]. Perhaps the sex-specific insulin response to food intake is a common phenomenon for different species: Men also have significantly higher postprandial insulin levels compared to women [19]. 

In the liver, refeeding caused multiple increase in *Fasn* mRNA level in males but did not affect its expression in females. In addition, the response from genes involved in gluconeogenesis was also more pronounced in males. Sex differences in liver *Fasn* expression after refeeding can be explained by the influence of insulin and leptin: Insulin stimulates *Fasn* expression, and leptin inhibits the stimulating effect of insulin (in rat hepatocytes) [38]. It is possible that in females, elevated leptin inhibits the stimulating effect of insulin on *Fasn* expression, thereby attenuating the increase in the expression of this gene during refeeding. The more pronounced inhibitory effect of refeeding on the expression of gluconeogenic genes in males may also be associated with increased insulin levels, since insulin inhibits *G6pc* and *Pck1* expression [39]. Male-specific increase in expression of hepatic *Fasn* in refed mice suggests that the rate of refeeding-induced lipogenesis was higher in males; however, additional studies are required to verify this assumption and to determine the physiological significance of this effect of refeeding.

*Cpt1a* expression in adipose tissue of different localizations responded to refeeding differently in males and females: It decreased in abdominal fat in males and in subcutaneous fat in females. Among studied genes in adipose tissue of different localizations, only *Cpt1a* demonstrated sex dimorphism in response to fasting/refeeding. This may indicate the impact of beta oxidation on sex differences in loss and recovery of adipose tissue of different localizations in response to fasting/refeeding and suggest that *Cpt1a* sensitivity to regulatory factors is sex-dependent and different in subcutaneous and abdominal adipose tissues.

In this work, we showed that refeeding activated *Fgf21* expression in VAT, BAT, and muscles, and such activation in VAT and muscles occurred in females only, and it was more pronounced in females than in males in BAT. These results are in agreement with our previous findings [16] and with the data of other authors [40]. In refed females, the increase in the local *Fgf21* mRNA expression levels in extra-hepatic tissues did not eliminate the refeeding-induced drop of the plasma FGF21 level, suggesting that FGF21 acts locally in an autocrine or paracrine manner. The mechanisms of the female specific upregulation of *Fgf21* in these tissues at refeeding are obscure. Agonists of PPARγ enhance *Fgf21* expression in perigonadal WAT [41] and agonists of PPARα and PPARγ in BAT [20]. In our experiment, expression rates of the *Fgf21* gene in refed female mice were not associated with PPAR expressions. In VAT, refeeding significantly increased *Pparg* expression only in males and *Fgf21* expression only in females. In BAT, refeeding induced opposite changes in the expression of *Pparg* and *Fgf21*: It inhibited *Pparg* expression and induced *Fgf21* expression. It has been shown that *Fgf21* gene expression is upregulated by hyperinsulinemia in muscle tissue [42], but refeeding induced muscle *Fgf21* expression in females but not in hyperinsulinemic males. 

It should be noted that in extra-hepatic tissues, the refeeding-induced female-specific increase in *Fgf21* gene expression was not associated with the upregulation of genes involved in carbohydrate or lipid oxidation. The local functions that may be regulated by FGF21 during refeeding are unclear. Still, the exact mechanisms underlying the sex differences in VAT, BAT, and muscle transcriptional responses to refeeding and the physiological significance of the elevated *Fgf21* gene expression require further investigation. 

The limitation of our work might be a possible heterogeneity of hormone and mRNA levels in female mice due to variations in sex steroid hormone concentrations during the estrus cycle. However, as concerns somatic cells, and especially hepatocytes, the pattern of growth hormone secretion and the level of estradiol receptors—which are sex-related and thus dramatically differ between males and females—are the major determinants of metabolic regulations, rather than the plasma concentrations of sex steroid hormones [43]. Besides, the heterogeneity of the measured parameters (as estimated by SD) was comparable in females and males. Thus, we suggest that, in the present experiment, variations in estrus cycle in the concentration of estradiol may have marginally affected the transcriptional response to fasting/refeeding when compared to the global effect of sex-related hormonal status. 

## 5. Conclusions

We found sex asymmetry both in the hormonal and transcriptional responses to states differed by nutrient availability. Refeeding-induced male-specific hyperinsulinemia was accompanied by elevation of mRNA level of *Fasn* that regulates the rate of lipogenesis in the liver. These results emphasize the role of insulin in the manifestation of sex differences in metabolic response to refeeding. A new finding of our study demonstrated that mRNA levels of *Cpt1*, which regulates the rate of fatty acid oxidation, showed sexual dimorphism in liver, muscles, and adipose tissues of different localization in response to fasting–refeeding. Fasting-induced sex-specific differences in liver *Fgf21* gene expression and the circulating FGF21 levels were associated with the upregulation of expression of genes involved in lipid oxidation in the liver (*Cpt1a*) and muscle (*Cpt1b, Ucp3*) in females. Refeeding-induced female-specific increase in *Fgf21* gene expression in extra-hepatic tissues (SAT, BAT, muscles) was not associated with the upregulation of these genes. These findings suggest that in mice, one of the molecular mechanisms underlying the sex asymmetric response to fasting/refeeding is *Fgf21* expression. It was biased toward females at both fasting and refeeding. Although the results of our work cannot be directly approximated to humans, they demonstrate the need to explore the role of FGF21 and insulin in adaptation to fasting/refeeding in men and women.

## Figures and Tables

**Figure 1 cells-08-01529-f001:**
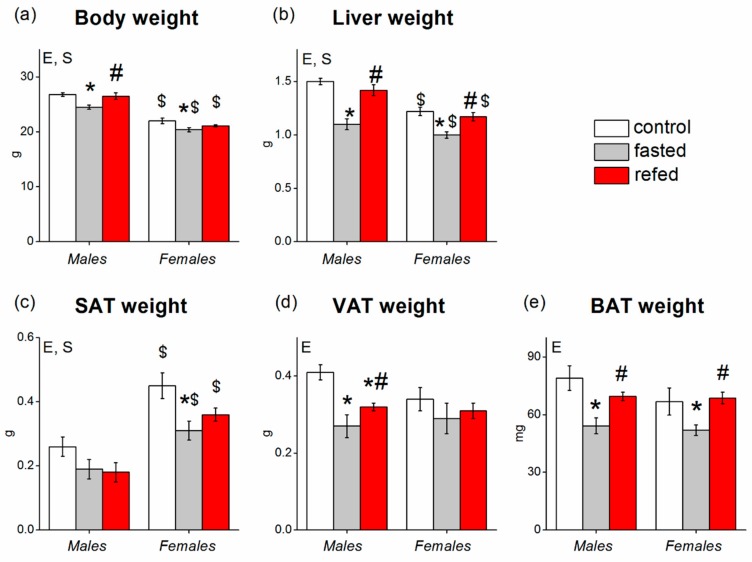
Weight of (**a**) body, (**b**) liver, (**c**) white subcutaneous adipose tissue (SAT), (**d**) white visceral adipose tissue (VAT), and (**e**) brown adipose tissue (BAT) in C57Bl/6J mice of three groups: control (fed ad libitum; white color), fasted for 24 h (grey color), and refed for 6 h after 24 h of fasting (red color). Two-way ANOVA was used with the factors of “sex” and “experimental group” (control, fasting, refeeding) with multiple comparisons using post hoc Fisher’s Least Significant Difference (LSD) test. Significance was determined as *p* < 0.05. S, sex effect; E, experimental group effect. * *p* < 0.05 versus control group, ^#^
*p* < 0.05 versus fasting group, ^$^
*p* < 0.05 - versus males in the same group.

**Figure 2 cells-08-01529-f002:**
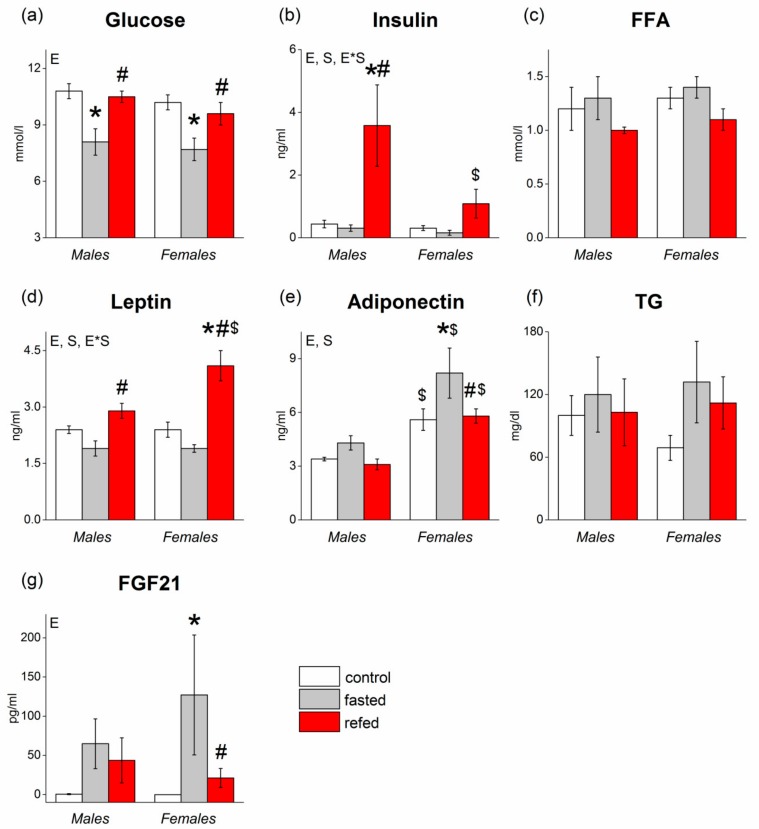
Concentrations of metabolites, including (**a**) glucose, (**c**) free fatty acid (FFA), (**f**) triglyceride (TG), and hormones, including (**b**) insulin, (**d**) leptin, (**e**) adiponectin, (**g**) FGF21, in the plasma of C57Bl/6J mice of three groups: control (fed ad libitum; white color), fasted for 24 h (grey color), and refed for 6 h after 24 h of fasting (red color). Two-way ANOVA was used with the factors of “sex” and “experimental group” (control, fasting, refeeding) with multiple comparisons using post hoc Fisher’s LSD test. Significance was determined as *p* < 0.05. S, sex effect; E, experimental group effect; and E*S, interactive effect of sex and experimental group. * *p* < 0.05 versus control group, ^#^
*p* < 0.05 versus fasting group, ^$^
*p* < 0.05 versus males in the same group.

**Figure 3 cells-08-01529-f003:**
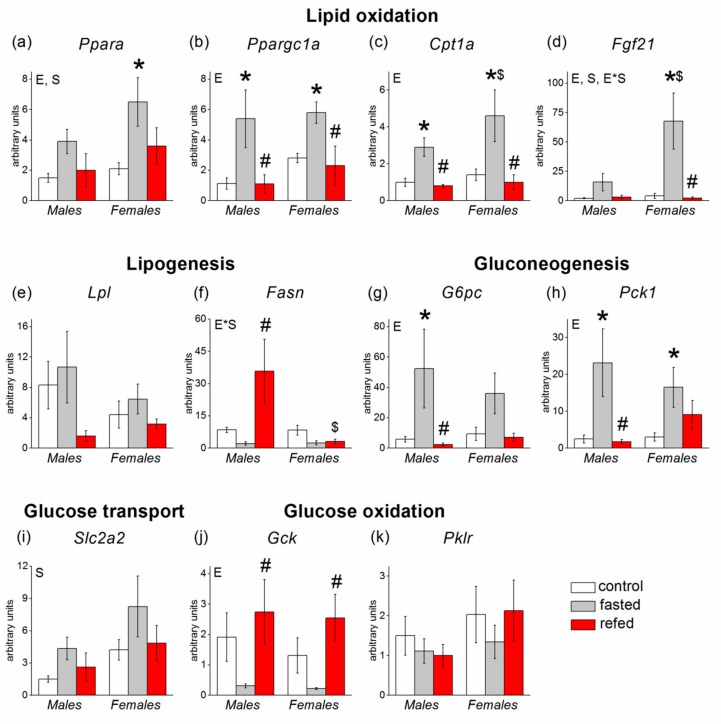
Expression of hepatic genes involved in glucose and lipid metabolism in C57Bl/6J mice of three groups: control (fed ad libitum; white color), fasted for 24 h (grey color), and refed for 6 h after 24 h of fasting (red color). (**a**) *Ppara*, (**b**) *Ppargc1a*, (**c**) *Cpt1a*, (**d**) *Fgf21*, (**e**) *Lpl*, (**f**) *Fasn*, (**g**) *G6pc*, (**h**) *Pck1*, (**i**) *Slc2a2*, (**j**) *Gck*, (**k**) *Pklr*. Two-way ANOVA was used with the factors of “sex” and “experimental group” (control, fasting, refeeding) with multiple comparisons using post hoc Fisher’s LSD test. Significance was determined as *p* < 0.05. S, sex effect; E, experimental group effect; and E*S, interactive effect of sex and experimental group. * *p* < 0.05 versus control group, ^#^
*p* < 0.05 versus fasting group, ^$^
*p* < 0.05 versus males in the same group.

**Figure 4 cells-08-01529-f004:**
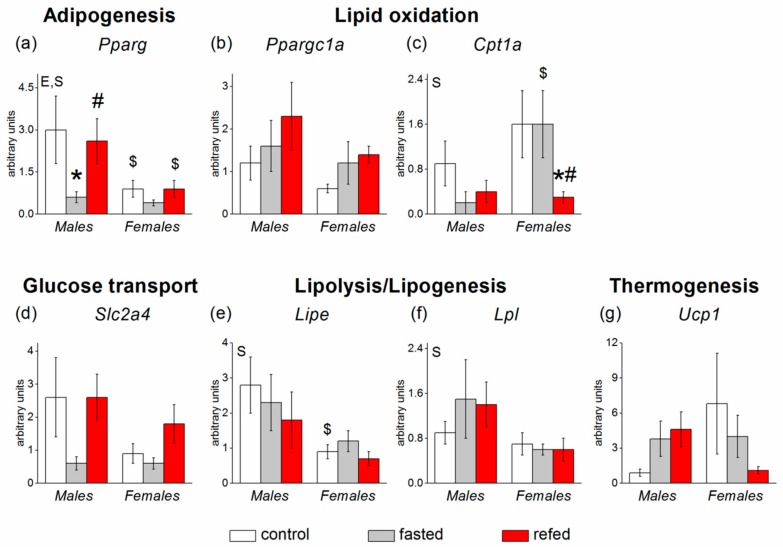
Expression of white subcutaneous adipose tissue (SAT) genes involved in glucose and lipid metabolism in C57Bl/6J mice of three groups: control (fed ad libitum; white color), fasted for 24 h (grey color), and refed for 6 h after 24 h of fasting (red color). (**a**) *Pparg*, (**b**) *Ppargc1a*, (**c**) *Cpt1b*, (**d**) *Slc2a4*, (**e**) *Lipe*, (**f**) *Lpl*, (**g**) *Ucp1*. Two-way ANOVA was used with the factors of “sex” and “experimental group” (control, fasting, refeeding) with multiple comparisons using post hoc Fisher’s LSD test. Significance was determined as *p* < 0.05. S, sex effect; E, experimental group effect. * *p* < 0.05 versus control group, ^#^
*p* < 0.05 versus fasting group, ^$^
*p* < 0.05 versus males in the same group.

**Figure 5 cells-08-01529-f005:**
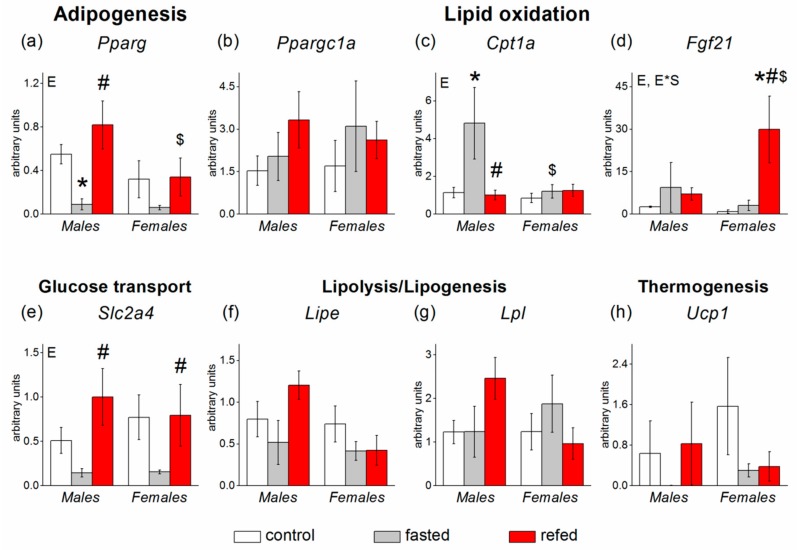
Expression of white visceral adipose tissue (VAT) genes involved in glucose and lipid metabolism in C57Bl/6J mice of three groups: control (fed ad libitum; white color), fasted for 24 h (grey color), and refed for 6 h after 24 h-fasting (red color). (**a**) *Pparg*, (**b**) *Ppargc1a*, (**c**) *Cpt1b*, (**d**) *Fgf21*, (**e**) *Slc2a4*, (**f**) *Lipe*, (**g**) *Lpl*, (**h**) *Ucp1*. Two-way ANOVA was used with the factors of “sex” and “experimental group” (control, fasting, refeeding) with multiple comparisons using post hoc Fisher’s LSD test. Significance was determined as *p* < 0.05. S, sex effect; E, experimental group effect; and E*S, interactive effect of sex and experimental group. * *p* < 0.05 versus control group, ^#^
*p* < 0.05 versus fasting group, ^$^
*p* < 0.05 versus males in the same group.

**Figure 6 cells-08-01529-f006:**
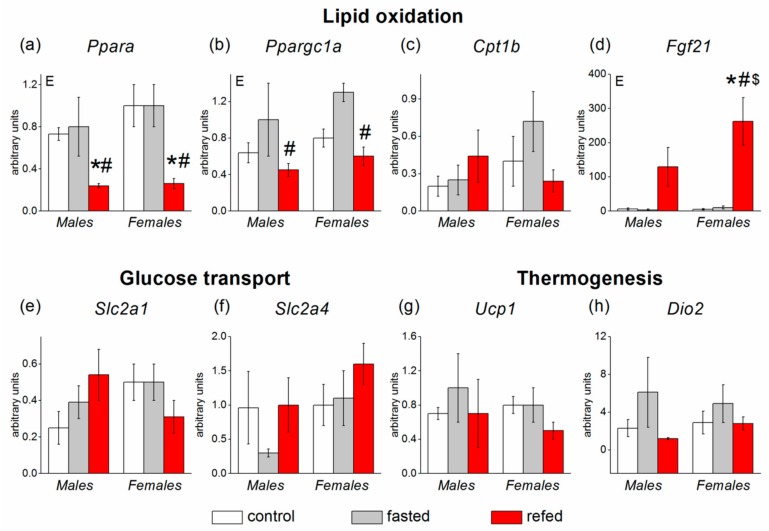
Expression of brown adipose tissue (BAT) genes involved in glucose and lipid metabolism in C57Bl/6J mice of three groups: control (fed ad libitum; white color), fasted for 24 h (grey color), and refed for 6 h after 24 h of fasting (red color). (**a**) *Ppara*, (**b**) *Ppargc1a*, (**c**) *Cpt1b*, (**d**) *Fgf21*, (**e**) *Slc2a1*, (**f**) *Slc2a4*, (**g**) *Ucp1*, (**h**) *Dio2*. Two-way ANOVA was used with the factors of “sex” and “experimental group” (control, fasting, refeeding) with multiple comparisons using post hoc Fisher’s LSD test. Significance was determined as *p* < 0.05. E, experimental group effect. * *p* < 0.05 versus control group, ^#^
*p* < 0.05 versus fasting group, ^$^
*p* < 0.05 versus males in the same group.

**Figure 7 cells-08-01529-f007:**
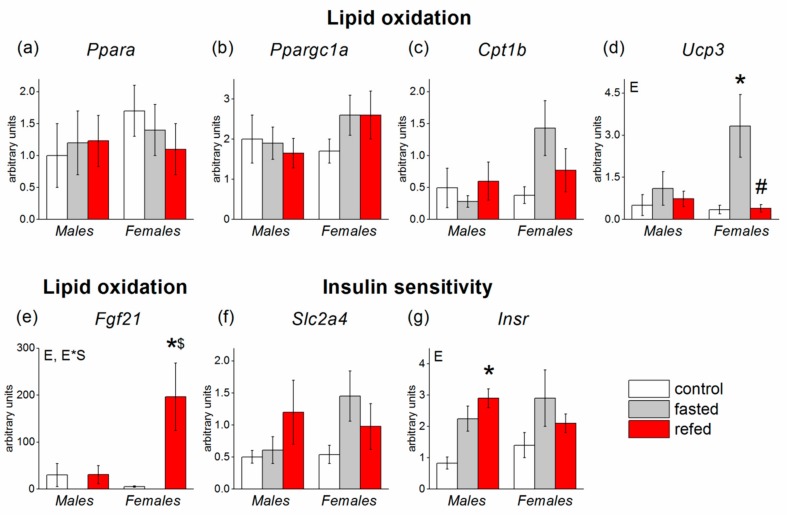
Expression of muscle genes involved in glucose and lipid metabolism in C57Bl/6J mice of three groups: control (fed ad libitum; white color), fasted for 24 h (grey color), and refed for 6 h after 24h-fasting (red color). (**a**) *Ppara*, (**b**) *Ppargc1a*, (**c**) *Cpt1b*, (**d**) *Ucp3*, (**e**) *Fgf21*, (**f**) *Slc2a4*, (**g**) *Insr*. mRNA levels of all genes were analyzed by two-way with the factors of “sex” and “experimental group” (control, fasting, refeeding—for all genes, for FGF21—only control and refeeding) with multiple comparisons using post hoc Fisher’s LSD test. Significance was determined as *p* < 0.05. E, experimental group effect; and E*S, interactive effect of sex and experimental group. * *p* < 0.05 versus control group, ^#^
*p*< 0.05 versus fasting group, ^$^
*p* < 0.05 versus males in the same group.

**Table 1 cells-08-01529-t001:** TaqMan gene expression assays.

Protein	Gene	Gene Expression Assay
Carnitine palmitoyltransferase 1a	*Cpt1a*	Mm01231183_m1
Carnitine palmitoyltransferase 1b	*Cpt1b*	Mm00487191_g1
Deiodinase, iodothyronine, type II	*Dio2*	Mm00515664_m1
Fatty acid synthase	*Fasn*	Mm00662319_m1
Fibroblast growth factor 21	*Fgf21*	Mm00840165_g1
Glucose-6-phosphatase, catalytic	*G6pc*	Mm00839363_m1
Glucokinase	*Gck*	Mm00439129_m1
Insulin receptor	*Insr*	Mm01211875_m1
Lipase, hormone sensitive	*Lipe*	Mm00495359_m1
Lipoprotein lipase	*Lpl*	Mm00434764_m1
Peroxisome proliferative activated receptor, gamma, coactivator 1 alpha	*Ppargc1a*	Mm01208835_m1
Peroxisome proliferator activated receptor alpha	*Ppara*	Mm0040939_m1
Peroxisome proliferator activated receptor gamma	*Pparg*	Mm00440940_m1
Phosphoenolpyruvate carboxykinase 1, cytosolic	*Pck1*	Mm01247058_m1
Pyruvate kinase liver and red blood cell	*Pklr*	Mm00443090_m1
Solute carrier family 2 (facilitated glucose transporter), member 1 (GLUT1)	*Slc2a1*	Mm00441480_m1
Solute carrier family 2 (facilitated glucose transporter), member 2 (GLUT2)	*Slc2a2*	Mm00446229_m1
Solute carrier family 2 (facilitated glucose transporter), member 4 (GLUT4)	*Slc2a4*	Mm00436615_m1
Uncoupling protein 1 (mitochondrial, proton carrier)	*Ucp1*	Mm01244861_m1
Uncoupling protein 3 (mitochondrial, proton carrier)	*Ucp3*	Mm01163394_m1
Beta-actin	*Actb*	Mm00607939_s1

**Table 2 cells-08-01529-t002:** Weight indexes of the liver and fat tissues in control, fasted and refed C57Bl/6J male and female mice.

	Males			Females			P
	Control	Fasting	Refeeding	Control	Fasting	Refeeding	
Liver (%)	5.58 ± 0.11 (*n* = 10)	4.47 ± 0.21 (*n* = 9) *	5.32 ± 0.11 (*n* = 9) ^#^	5.47 ± 0.11 (*n* = 10)	4.90 ± 0.13 (*n* = 8) *^,$^	5.54 ± 0.14 (*n* = 9) ^#^	E
SAT (%)	0.98 ± 0.11 (*n* = 10)	0.76 ± 0.13 (*n* = 9)	0.68 ± 0.10 (*n* = 9)	1.99 ± 0.18 (*n* = 10) ^$^	1.48 ± 0.13 (*n* = 8) *^,$^	1.71 ± 0.11 (*n* = 9) ^$^	S
VAT (%)	1.52 ± 0.07 (*n* = 10)	1.10 ± 0.13 (*n* = 9)	1.20 ± 0.05 (*n* = 9)	1.52 ± 0.13 (*n* = 10)	1.39 ± 0.21 (*n* = 8)	1.48 ± 0.11 (*n* = 9)	
BAT (%)	0.29 ± 0.02 (*n* = 10)	0.22 ± 0.02 (*n* = 9) *	0.26 ± 0.01 (*n* = 9)	0.30 ± 0.03 (*n* = 10)	0.25 ± 0.01 (*n* = 7)	0.33 ± 0.02 (*n* = 9) ^#,$^	E, S

Abbreviations: SAT–subcutaneous adipose tissue, VAT–visceral adipose tissue, BAT–brown adipose tissue, BW–body weight. The SAT, VAT, BAT, and liver mass indexes were calculated as the ratio of tissue mass to body weight expressed as a percentage. Two-way ANOVA was used with the factors “sex” and “experimental group” (control, fasting, refeeding) with multiple comparisons using post hoc Fisher’s Least Significant Difference (LSD) test. Significance was determined as *p* < 0.05. S, sex effect; E, experimental group effect. * *p* < 0.05 versus control group, ^#^
*p* < 0.05 versus fasting group, ^$^
*p* < 0.05—versus males in the same group.

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
