# Peer review of "Sex Differences in Liver, Adipose Tissue, and Muscle Transcriptional Response to Fasting and Refeeding in Mice"

_cells, 2019, doi:10.3390/cells8121529_

Round 1

Reviewer 1 Report

It is of great research interest in understanding sex differences in metabolic regulation and developing tailored therapeutic strategies to each sex. The manuscript “SEX DIFFERENCES IN LIVER, ADIPOSE TISSUE AND MUSCLE TRANSCRIPTIONAL RESPONSE TO FASTING AND REFEEDING IN MICE”, authored by Bazhan et al. investigated sex different hormonal responses and transcriptional responses in liver, white and brown adipose, and muscle tissues to 24-hour fasting and followed by 6-hour refeeding. The authors reported greater changes in circulating levels of fibroblast growth factor-21 (FGF21) and adipokines, as well as expression of Fgf21 and other genes involved in fatty acid oxidation, in females than males. This is an important study due to dieting as one of the most common body weight control methods and yoyo dieting seen in both men and women. This submission is improved, and has incorporated many of the previous comments and suggestions.

One of the previous major concerns was that, although this study aimed to study sex differences by comparing female and male mice, sex hormones were not measured, and estrous phase of female mice was not monitored either.  There is concern that females might not display their regular estrous cycles and had reduced estrogen levels during fasting. This comment was not addressed in this submission. The author should at least discuss this concern.

I have two specific comments.

(1) It is correct to say “mRNA level” and “gene expression”, as seen at many places in the manuscript, but it is not appropriate to say “mRNA expression” that appears at many places in the manuscript.

(2) Line 124: adipose tissue at paragonadal location was the visceral adipose tissue (VAT) measured in this study.  The paragonadal adipose tissue of male and female rodents are quite different. Male gonadal adipose tissue is located in the scrotum, not in the abdominal cavity; whereas female gonadal adipose tissue is located in the abdominal cavity. Therefore, author need to discuss whether it is appropriate to choose these adipose tissues as VAT.  The authors also need to discuss if musculus quadriceps femoris is oxidative or glycolytic muscle, and if it is a good choice to study energy metabolism.

Author Response

We are gratefull to all reviewers for their comments. Thanks to these comments, we radically revised our paper. In addition, we found several errors that are corrected in this version of the manuscript.

We found an error in the calculation of the relative weights of organs and tissues in refed mice (the weights of organs were normalized to the body weights before refeeding). In the present version, this error is corrected, the figure with weight indices is replaced by the figure with absolute weights (Figure 1), and relative weights are presented in Table 2.

In Figures 4 and 5, Cpt1 gene was marked incorrectly, this error was corrected. In addition, we removed from the figures the marks of statistically reliable differences confirmed by Students t-test .

The experimental groups in which weights and blood hormones levels were estimated consisted of 8-10 animals in each group. To assess gene expression, groups were reduced to 6-7 animals. These numbers (6-7 animals) were incorrectly indicated for weight and hormonal characteristics.

We have completely revised discussion and abstract, and partially, results.

One of the previous major concerns was that, although this study aimed to study sex differences by comparing female and male mice, sex hormones were not measured, and estrous phase of female mice was not monitored either.  There is concern that females might not display their regular estrous cycles and had reduced estrogen levels during fasting. This comment was not

Unfortunately, for technical reasons (estradiol kit turned out to be of poor quality), we failed to measure the plasma estradiol concentrations in the experimental groups. At the end of Discussion, we included paragraph concerning a possible influence of estradiol on gene expression in cycling female mice.

It is correct to say “mRNA level” and “gene expression”, as seen at many places in the manuscript, but it is not appropriate to say “mRNA expression” that appears at many places in the manuscript.

It was corrected.

(2) Line 124: adipose tissue at paragonadal location was the visceral adipose tissue (VAT) measured in this study.  The paragonadal adipose tissue of male and female rodents are quite different. Male gonadal adipose tissue is located in the scrotum, not in the abdominal cavity; whereas female gonadal adipose tissue is located in the abdominal cavity. Therefore, author need to discuss whether it is appropriate to choose these adipose tissues as VAT.  

Rodents harbor visceral fat pads in the perigonadal region, known as epididymal in males and periovarian in females.  The perigonadal fat pad in male and female rodents, in contrast to humans, is considered as visceral fat (Cinti et al., 2005).  

Cinti S. The adipose organ. Prostaglandins Leukot Essent Fatty Acids. 2005 Jul;73(1):9-15. Review. PubMed PMID: 15936182.

Item F, Konrad D. Visceral fat and metabolic inflammation: the portal theory revisited. Obes Rev. 2012 Dec;13 Suppl 2:30-9. doi: 10.1111/j.1467-789X.2012.01035.x.

The authors also need to discuss if musculus quadriceps femoris is oxidative or glycolytic muscle, and if it is a good choice to study energy metabolism.

The following text was added to the Materials and Methods. «In mice, musculus quadriceps femoris is characterized by combination of two metabolic pathways: anaerobic glycolysis and aerobic beta-oxidation, which closely interact with each other (Eimre et al., 2018).  Therefore, the transcription of genes involved in glycolysis and in the fatty acid oxidation was measured in our study.»

Eimre M, Paju K, Peet N, Kadaja L, Tarrend M, Kasvandik S, Seppet J, Ivask M, Orlova E, Kõks S. Increased Mitochondrial Protein Levels and Bioenergetics in the Musculus Rectus Femoris of Wfs1-Deficient Mice. Oxid Med Cell Longev. 2018 Nov 21;2018:3175313. doi: 10.1155/2018/3175313.  

Reviewer 2 Report

The manuscript entitled “Sex differences in liver, adipose tissue and muscle transcriptional response to fasting and refeeding in mice” aims to evaluate hormonal and transcriptional responses to fasting and refeeding in male and female C57BL6 mice.

The manuscript is well-structured, the flow of information is correct, and the experimental approach is interesting. Research in metabolism has been always done with males and low studies using females are available.

The aim of the manuscript presented by Bazhan et al. is significant and relevant, but some issues make it non-acceptable for publication.

1.- Lines 69-87 require structure. It is not clear and difficult to understand where and why the authors measure some specific genes in some tissues. In some cases, the authors have included lot of information about a gene/enzyme but for other cases they have only added the name of the gene. I missed some references that should be added and some statements explaining why the authors decided these genes, the relevance of these ones for the aim of the paper.For instance, the authors should discuss why do they measure Ucp3. The expression levels of this gene increase when fatty acid supplies to mitochondria exceed their oxidation capacity and the protein enables the export of fatty acids from mitochondria.The authors should introduce a more accurate description of the metabolic response and metabolic pathways activated/repressed in the fast, fed or refed state.

2.- Lines 69-87: the authors mentioned some genes that lately are analyzed but no others. All should be described in the context of the manuscript to understand their role in fast, fed or refed metabolic response.

3.- Please check the symbols of the genes in all the manuscript. For instance, some PPARs are as PPARa and others as PPARa.

4.- line 104: the authors should identify correctly the mice strain. C57BL is not the complete name.

5.- Lines 125: the authors just explain how they have calculated the adipose tissue index but they have not included the rest.  

6.- Figure 1.- As the authors weighted the tissues, they should represent the tissue weight instead of % of BW. Can the body weight reduction in fasting state or the increase in refeeding be explained by the changes detected in the tissues the authors analyze?

7.- How do the authors explain that 24h of fasting did not change the FFA levels or the TG neither in males nor females? Or that the FGF21 levels were not induced in fasted males?

8.- Line 205: the authors should erase the text [lewitt01 and Lewit02]

9.- Figure 4: Why the SAT is the only tissue where the authors did not measure Fgf21?

10.- The authors should explain why the fasted stated did not affect the Cpt1b expression neither in adipose tissues nor in skeletal muscle in males and in some cases, it is down regulated.

11.- Discussion is largely speculative with many assumptions that are not support by the results. Most of the results from the paper are not conclusive, just some changes are significant and there is not enough data to draw the complete picture. Some examples:

- Line 405: the authors suggest that in females FFA oxidation may lead to formation of ketone bodies. The authors can confirm this by measuring the ketone bodies and some genes involved in ketogenesis such as the hydroxy-methylglutaryl CoA synthase 2 (HMGCS2).

- The authors mention several times the differences in insulin sensitivity between females and males as a key point to explain their results. The authors should perform and ITT or GTT to conform this hypothesis.

- The authors assume that the intracellular glucose levels that can be at least in part the responsible of the FGF21 induction would be higher in females. They should demonstrate this and also check the mRNA levels of ChREBP.

The authors should revise the discussion and make it in accordance to their results.  The data are overinterpreted The authors should deepen in some of the assumptions or suggestions they mention to increase the relevance of the paper.

12.- Discussion: Some references used did not say what the authors write in the text. For instance, Hondares et al [55] describes the effect of fats from milk in FGF21 induction and thermogenesis activation in mice puppies but does not talk about the re-accumulation of glycogen stores after cold exposure.

Author Response

We are gratefull to all reviewers for their comments. Thanks to these comments, we radically revised our paper. In addition, we found several errors that are corrected in this version of the manuscript.

We found an error in the calculation of the relative weights of organs and tissues in refed mice (the weights of organs were normalized to the body weights before refeeding). In the present version, this error is corrected, the figure with weight indices is replaced by the figure with absolute weights (Figure 1), and relative weights are presented in Table 2.

In Figures 4 and 5, Cpt1 gene was marked incorrectly, this error was corrected. In addition, we removed from the figures the marks of statistically reliable differences confirmed by Students t-test .

The experimental groups in which weights and blood hormones levels were estimated consisted of 8-10 animals per group. To assess gene expression, groups were reduced to 6-7 animals. These numbers (6-7 animals) were incorrectly indicated for weight and hormonal characteristics.

We have completely revised discussion and abstract, and partially, results.

1.- Lines 69-87 require structure. It is not clear and difficult to understand where and why the authors measure some specific genes in some tissues. In some cases, the authors have included lot of information about a gene/enzyme but for other cases they have only added the name of the gene. I missed some references that should be added and some statements explaining why the authors decided these genes, the relevance of these ones for the aim of the paper. For instance, the authors should discuss why do they measure Ucp3. The expression levels of this gene increase when fatty acid supplies to mitochondria exceed their oxidation capacity and the protein enables the export of fatty acids from mitochondria. The authors should introduce a more accurate description of the metabolic response and metabolic pathways activated/repressed in the fast, fed or refed state.

2.- Lines 69-87: the authors mentioned some genes that lately are analyzed but no others.

The text was rewritten.

Fatty acid oxidation, ketogenesis, gluconeogenesis in the liver, lipolysis of triglycerides in white adipose tissue, thermogenesis in brown adipose tissue, fatty acid, and glucose oxidation in extrahepatic metabolic organs (WAT, BAT, muscles) are the main processes involved in adaptation to the cycle fasting-refeeding.  We have focused on the mRNA levels of Fgf21  and ligand-activated transcription factors (PPARalpha, and PPARgamma) involved in regulation of FGF21 expression [10,20].  Among the genes related to lipid metabolism, we measured the expression of genes involved in the fatty acid oxidation (peroxisome proliferator activated receptor γ coactivator protein-1α, Ppargc1a, carnitine palmitoyltransferase 1a Cpt1α, and uncoupling protein 3, Ucp3) also genes controlling lipolysis (hormone-sensitive lipase, Lipe) and lipogenesis (fatty acid synthase, Fas, and lipoprotein lipase, Lpl, the enzyme that hydrolyzes blood triglycerides and promotes the cellular uptake of free fatty acids [21]).  In WAT, PPARγ and glucose transporter 4 (Slc2a4) genes are also considered as lipogenic genes [22]. Among the genes related to glucose turnover we measured hepatic expression of gluconeogenic genes (glucose-6-phosphatase, G6p, and phosphoenolpyruvate carboxykinase 1, Pck1), genes involved in glucose oxidation (glucokinase, Gck, and pyruvate kinase, Pklr), and glucose transporter 2, Slc2a2. In adipose tissues and muscle, we measured mRNA level of Slc2a1, Slc2a4, and insulin receptor gene, Insr, the latter – only in the muscle. Among thermogenic genes we measured expression of uncoupling protein 1, Ucp1, and deiodinase iodothyronine 2, Dio2, the latter – only in the brown adipose tissue

3.- Please check the symbols of the genes in all the manuscript. For instance, some PPARs are as PPARa and others as PPARa.

It was done.

4.- line 104: the authors should identify correctly the mice strain. C57BL is not the complete name.

It was done.

5.- Lines 125: the authors just explain how they have calculated the adipose tissue index but they have not included the rest.  

In addition to adipose tissue indexes, a liver index was also calculated. Relevant additions have been made to the text.

6.- Figure 1.- As the authors weighted the tissues, they should represent the tissue weight instead of % of BW. Can the body weight reduction in fasting state or the increase in refeeding be explained by the changes detected in the tissues the authors analyze?

We agree that the absolute weights of organs and tissues may better illustrate the changes that fasting and refeeding cause. Therefore, we replaced the figure with relative weights with the figure with absolute weights (Figure 1), and presented the indices in the table 2.

Of course, BW reduction and recovery during fasting/refeeding can not be explained by the changes in the organ weights. We think that the changes in the body weight induced by 24-h food deprivation and subsequent refeeding are mainly due to bowel depletion during the fasting period and food consumed during the refeeding period. In males, BW gain during refeeding was about 1,9 g, and they consumed about 1,9 g of food.

7.- How do the authors explain that 24h of fasting did not change the FFA levels or the TG neither in males nor females? Or that the FGF21 levels were not induced in fasted males?

In the WAT, free fatty acid release (i.e., lipolysis) in response to fasting, is a critical step aimed at maintaining whole body energy homeostasis. Fasting-induced hepatic steatosis results from repartitioning of FFA released from the adipose tissue to liver.  In the liver, FFAs can either be used for beta-oxidation in mitochondria or reesterified into TG. TG can be stored or secreted as VLDL. In turn, TG-rich VLDL particles are lipolyzed by LPL and deliver FFAs to other tissues, such as skeletal muscle, where FFAs are used for beta-oxidation.  If above processes proceed with equal intensity plasma FFA and TG levels in fasting mice will not differ from those in control mice.  Some authors also did not find acute fasting-induced increase in FFA and TG plasma levels in WT mice (Heijboer et al., 2005, Hotta et al., 2009, Inagaki et al., 2007, Li et al., 2014, Sanchez et al., 2009).

Heijboer, A. C., E. Donga, P. J. Voshol, Z-C. Dang, L. M. Havekes, J. A. Romijn, and E. P. M. Corssmit. Sixteen hours of fasting differentially affects hepatic and muscle insulin sensitivity in mice. J. Lipid Res. 2005. 46: 582–588.

Yuhei Hotta,* Hirotoshi Nakamura,* Morichika Konishi, Yusuke Murata, Hiroyuki Takagi, Shigenobu Matsumura, Kazuo Inoue, Tohru Fushiki, and Nobuyuki Itoh Fibroblast Growth Factor 21 Regulates Lipolysis in

Sanchez J, Palou A, Pico C. 2009. Response to carbohydrate and fat refeeding in the expression of genes involved in nutrient partitioning and metabolism: striking effects on fibroblast growth factor-21 induction. Endocrinology 150:5341–50White Adipose Tissue But Is Not Required for Ketogenesis and Triglyceride Clearance in Liver Endocrinology 150: 4625–4633, 2009

Or that the FGF21 levels were not induced in fasted males?

The Results section has been rewritten according to the comments of reviewers. 

2-way ANOVA revealed effect of experimental group so we wrote that “Fasting also sharply increased and refeeding decreased circulating FGF21 levels (P<0.05) both in males and females.” Consequently, fasting induced plasma Fgf21 levels in males, in agreement with the results of other researchers (Li et al., 2014).

8.- Line 205: the authors should erase the text [lewitt01 and Lewit02]

It was done.

9.- Figure 4: Why the SAT is the only tissue where the authors did not measure Fgf21?

We tried to measure Fgf21 mRNA levels in SAT along with other tissues, but they were so low that we could not measure them in mice of both sexes and all experimental groups. Now, it is indicated in Results.

10.- The authors should explain why the fasted stated did not affect the Cpt1b expression neither in adipose tissues nor in skeletal muscle in males and in some cases, it is down regulated.

A mistake was made in the text of the article: the expression of the Cpt1α gene, not Cpt1β gene, was measured in white adipose tissue. Our data demonstrated that fasting-induced regulation of Cpt1α and Cpt1β transcription was tissue- and sex-specific. Fasting upregulated and refeeding downregulated Cpt1α mRNA levels only in pgWAT of males and Cpt1β mRNA levels only in muscle of females.  Palou et al. (2010) also demonstrated that Cpt1α expression increased after 24 h of fasting in the retroperitoneal not the subcutaneous depot in male rats. Since in males, fasting-induced changes in Cpt1β gene transcription in BAT and muscle were not statistically significant we refrained from discussing these issues.

Palou M, Sánchez J, Priego T, Rodríguez AM, Picó C, Palou A. Regional differences in the expression of genes involved in lipid metabolism in adipose tissue in response to short- and medium-term fasting and refeeding. J Nutr Biochem. 2010 Jan;21(1):23-33. doi: 10.1016/j.jnutbio.2008.10.001.

11.- Discussion is largely speculative with many assumptions that are not support by the results. Most of the results from the paper are not conclusive, just some changes are significant and there is not enough data to draw the complete picture. Some examples:

- Line 405: the authors suggest that in females FFA oxidation may lead to formation of ketone bodies. The authors can confirm this by measuring the ketone bodies and some genes involved in ketogenesis such as the hydroxy-methylglutaryl CoA synthase 2 (HMGCS2).

- The authors mention several times the differences in insulin sensitivity between females and males as a key point to explain their results. The authors should perform and ITT or GTT to conform this hypothesis.

- The authors assume that the intracellular glucose levels that can be at least in part the responsible of the FGF21 induction would be higher in females. They should demonstrate this and also check the mRNA levels of ChREBP.

The authors should revise the discussion and make it in accordance to their results.  The data are overinterpreted.  The authors should deepen in some of the assumptions or suggestions they mention to increase the relevance of the paper.

We agree that discussion of the article turned speculative and repetitive.  It has been radically changed and concentrated around the main objective of the study - sex differences in hormonal and transcriptional responses to fasting and refeeding in mice.  We removed some speculations that had no experimental verification, including those that you listed in paragraph 11.

12.- Discussion: Some references used did not say what the authors write in the text. For instance, Hondares et al [55] describes the effect of fats from milk in FGF21 induction and thermogenesis activation in mice puppies but does not talk about the re-accumulation of glycogen stores after cold exposure.

This was an erroneous citation.  The correct link is given below:

Hondares E, Iglesias R, Giralt A, Gonzalez FJ, Giralt M, et al. 2011. Thermogenic activation induces FGF21 expression and release in brown adipose tissue. J. Biol. Chem. 286:12983–90

Reviewer 3 Report

Bazhan and colleagues evaluate sex-dependent hormonal and transcriptional responses to fasting and refeeding in male and female mice. One of the molecular mechanisms underlying the sex asymmetric response to fasting/refeeding is FGF21 expression, that in turn regulates  the expression of genes and protein involved in lipid and glucose metabolism. The present study has certainly a good potential although it is not clearly described. In my opinion, the manuscript is not available for publication in Cells.

The mayor criticism regards the number of animals used for experiments. The authors declare that three experimental groups were used and that six to seven mice were in each experimental group. It is not specify the  number of male and female mice in each experimental group. For the reviser it is not clear how the authors have performed statistical analysis with this few number of animals! Other critical point: the description of results appears very confusing and, some times, the sentences are not corresponding to the graphs (lines 174-179; 214-218; etc ). At the same way Discussion, in some parts, appears confused, in some others, repetitive.

Others comments:

The English language have to be improve; usually it is not understandable what the authors want to say. What do you mean with “lipid carbohydrate”?(pag. 2 line 94, such as in others paragraphs) In the paragraph Study Design line 125, is described how has been calculated the adipose tissue mass index, bat the liver index is not specify. In the paragraph 3.1, line 170-171, you should indicates the groups of mice in which has been observed the result. In the paragraph 3.2 the reference to the figure (Figure 3) where the results are shown is missing.

The manuscript could be ameliorated also by measuring sex hormones concentration (such as estradiol) in each experimental group.

Author Response

We are gratefull to all reviewers for their comments. Thanks to these comments, we radically revised our paper. In addition, we found several errors that are corrected in this version of the manuscript.

We found an error in the calculation of the relative weights of organs and tissues in refed mice (the weights of organs were normalized to the body weights before refeeding). In the present version, this error is corrected, the figure with weight indices is replaced by the figure with absolute weights (Figure 1), and relative weights are presented in Table 2.

In Figures 4 and 5, Cpt1 gene was marked incorrectly, this error was corrected. In addition, we removed from the figures the marks of statistically reliable differences confirmed by Students t-test .

We have completely revised discussion and abstract, and partially, results.

The mayor criticism regards the number of animals used for experiments. The authors declare that three experimental groups were used and that six to seven mice were in each experimental group. It is not specify the  number of male and female mice in each experimental group. For the reviser it is not clear how the authors have performed statistical analysis with this few number of animals!

The experimental groups in which weights and blood hormones levels were estimated consisted of 8-10 animals in each group. To assess gene expression, groups were reduced to 6-7 animals. These numbers (6-7 animals) were incorrectly indicated for weight and hormonal characteristics. We indicated the number of mice in experimental groups in Methods.

Other critical point: the description of results appears very confusing and, some times, the sentences are not corresponding to the graphs (lines 174-179; 214-218; etc ). At the same way Discussion, in some parts, appears confused, in some others, repetitive.

We agree that Discussion turned speculative and repetitive.  It has been radically changed and concentrated around the main objective of the study - sex differences in hormonal and transcriptional responses to fasting and refeeding in mice. 

Others comments:

The English language have to be improve; usually it is not understandable what the authors want to say. What do you mean with “lipid carbohydrate”?(pag. 2 line 94, such as in others paragraphs)

An extra word “lipid” was mistakenly written in this sentence. It was removed.

In the paragraph Study Design line 125, is described how has been calculated the adipose tissue mass index, bat the liver index is not specify.

Relevant additions have been made to the text. “The SAT, VAT, BAT, and liver mass indexes were calculated as the ratio of tissue mass to body weight expressed as a percentage”.  Line 142

In the paragraph 3.1, line 170-171, you should indicates the groups of mice in which has been observed the result. In the paragraph 3.2 the reference to the figure (Figure 3) where the results are shown is missing.

Thank you very much for comments regarding the presentation of the results. The Results section has been rewritten according to the comments of reviewers. All errors you noted were corrected.

The manuscript could be ameliorated also by measuring sex hormones concentration (such as estradiol) in each experimental group.

Unfortunately, for technical reasons (estradiol kit turned out to be of poor quality), we failed to measure the plasma estradiol concentrations in the experimental groups. At the end of Discussion, we included paragraph concerning a possible influence of estradiol on gene expression in females during estrous cycle.

Round 2

Reviewer 2 Report

The authors clearly improved the quality of the paper in this last version. They make an effort to adjust the figures to a more comprehensive way and to rewrite the discussion according to their results and the previous published data. 

Author Response

We thank the Reviewer for the work and suggestions that were very important for us when we revised our manuscript

Reviewer 3 Report

I believe that the authors have well understand the criticism of the manuscript. It has been improved, so that I think that it can be accepted in the present form.

Author Response

We thank the Reviewer for the work with our manuscript and suggestions that were very important for us when we revised the manuscript